# Genetic, Epigenetic, and Steroidogenic Modulation Mechanisms in Endometriosis

**DOI:** 10.3390/jcm9051309

**Published:** 2020-05-02

**Authors:** Anna Zubrzycka, Marek Zubrzycki, Ewelina Perdas, Maria Zubrzycka

**Affiliations:** 1Department of Biomedicine and Genetics, Medical University of Lodz, Pomorska 251, 92-213 Lodz, Poland; Poland; azubrzycka@op.pl; 2Department of Operative and Conservative Gynecology, K. Jonscher Memorial Hospital, Milionowa 14, 93-113 Lodz, Poland; 3Department of Cardiac Surgery and Transplantology, The Cardinal Stefan Wyszynski Institute of Cardiology, Alpejska 42, 04-628 Warsaw, Poland; marek.zubrzycki@op.pl; 4Department of Cardiovascular Physiology, Faculty of Medicine, Medical University of Lodz, Mazowiecka 6/8, 92-215 Lodz, Poland; ewelina.perdas@umed.lodz.pl

**Keywords:** endometriosis, genetics, epigenetics modifications, DNA methylation, histone proteins, microRNA

## Abstract

Endometriosis is a chronic gynecological disease, affecting up to 10% of reproductive-age women. The exact cause of the disease is unknown; however, it is a heritable condition affected by multiple genetic, epigenetic, and environmental factors. Previous studies reported variations in the epigenetic patterns of numerous genes known to be involved in the aberrant modulation of cell cycle steroidogenesis, abnormal hormonal, immune and inflammatory status in endometriosis, apoptosis, adhesion, angiogenesis, proliferation, immune and inflammatory processes, response to hypoxia, steroidogenic pathway and hormone signaling are involved in the pathogenesis of endometriosis. Accumulating evidence suggest that various epigenetic aberrations may contribute to the pathogenesis of endometriosis. Among them, DNA methyltransferases, histone deacetylators, and non-coding microRNAs demonstrate differential expression within endometriotic lesions and in the endometrium of patients with endometriosis. It has been indicated that the identification of epigenetic differences within the DNA or histone proteins may contribute to the discovery of a useful prognostic biomarker, which could aid in the future earlier detection, timely diagnosis, and initiation of a new approach to the treatment of endometriosis, as well as inform us about the effectiveness of treatment and the stage of the disease. As the etiology of endometriosis is highly complex and still far from being fully elucidated, the presented review focuses on different approaches to identify the genetic and epigenetic links of endometriosis and its pathogenesis.

## 1. Introduction

Endometriosis, one of most common benign gynecologic disorders, is a chronic, inflammatory and estrogen-dependent disease, involving proliferation of endometrial and stromal tissue outside the uterine cavity [1]. This disease is diagnosed in approximately 10% of all reproductive-age women, with its prevalence increasing to 50% in infertile women [2]. Endometriosis affects more frequently women of Philippine, Indian, Japanese, and Korean origin [3]. Chronic pelvic pain, dysmenorrhea, and impaired fertility are the predominant symptoms, significantly affecting the quality of life of women with endometriosis [4]. Endometriosis can lead to such pathological processes as peritoneal inflammation, development of fibrosis, and ovarian cysts [5]. Despite the high prevalence of the disease, the diagnosis of endometriosis is often delayed by 7–10 years due to the complexity of the pathogenesis, diversity of symptoms as well as the lack of a timely non-invasive diagnostic tool [6,7]. Laparoscopy is currently established as the gold standard for the definitive identification of endometriosis and histological biopsy as the method to confirm the diagnosis [1]. Transvaginal ultrasound should be the first-line investigation used for diagnostic purposes in patients with suspected endometriosis. Currently, magnetic resonance imaging (MRI) is an important addition to the non-invasive diagnosis of extraovarian endometriosis and should be performed before the institution of treatment, especially surgical one [8]. Biomarkers such as CA-125 have also been utilized for diagnosis, although they demonstrate low specificity. Molecular biomarkers, such as microRNA (miRNAs) have also been studied [9,10].

The cause of endometriosis remains unknown to date, and its complex etiopathogenesis has been only partially elucidated (Figure 1) [11,12]. Endometriosis is a multifactorial disease, generated by a combined action of multiple genetic, epigenetic, and environmental factors, all of them interacting with each other in order to yield the disease phenotype [13]. Previous genetic studies on endometriosis did not succeed in identification of the genetic variants strongly correlated with the risk of the disease. Nowadays, a better understanding of the genetic risk factors associated with endometriosis has been achieved owing to the use of advanced technological applications. Candidate gene studies, gene association and genome wide association studies (GWAS) have already yielded over 30 candidate genes [14]. However, studies aimed at determination of the usefulness of these genes for understanding the pathogenesis of endometriosis are still underway. To date, 27 independent single-nucleotide polymorphisms (SNPs) have been significantly associated with endometriosis upon GWAS, explaining over 5% of disease variances [15,16].

GWAS association studies conducted in various samples have allowed to identify the disease-susceptibility loci implicated in matrix remodeling, transcription regulation, cell cycle regulation and signaling, cell adhesion, inflammation, immunity, oxidative stress and steroid hormone receptors [4,17,18]. The literature data indicate the existence of a number of differences in the associations of the disease with the frequency of genetic polymorphisms in women from different ethnic groups [19].

Accumulating evidence supports the concept that endometriosis is a disease associated with an epigenetic disorder [17,20,21]. The factors that may affect the expression of the genes include both intracellular factors and environmental stimuli. At the cellular level, the epigenetic information is reflected first of all by the level of DNA methylation, histone modifications, and miRNA expression [22,23,24]. These epigenetic players are regulated microenvironmental cues, such as hypoxia, proinflammatory cytokines, and locally produced estradiol, and they reciprocally regulate the process, or the response to those stimuli.

A marked progress in this area has been achieved, mainly as a result of identification of new candidate genes and numerous SNPs closely related to endometriosis [25], of genetic and epigenetic mechanisms of its regulation [26,27], and of endometrial stem cells [28], as well as transcriptome and miRNA analyses of the endometrium and endometriotic cells [20,29].

This fact can be used for diagnostic purposes to detect the disease and to forecast its course. From the point of view of the therapy, it is important that new epigenetic markers can also be used to assess the effectiveness of the treatment undertaken. This prompts further exploration to find new epigenetic markers, especially for endometriosis, for which the current diagnostic methods are insufficient.

As the etiology of endometriosis is highly complex and still far from being fully elucidated, the current review aims to offer a comprehensive summary of the available evidence to identify the genetic and epigenetic links of endometriosis and its pathogenesis. We conducted a narrative review synthesizing the findings reported in the English literature retrieved from computerized MEDLINE database (accessed through PubMed) up until March 2020, using the keywords “endometriosis”, “genetic” and “epigenetic,” “DNA methylation,” “histone modification,” and “microRNA.”

## 2. Genetic Profile of Endometriosis

### 2.1. Familial Studies in Endometriosis

Familial studies aim to define inheritance trends. Endometriosis is considered to have a complex genetic etiology requiring the interaction of multiple genetic and environmental risk factors that contribute to formation of the disease phenotype. Endometriosis as a phenotype seems to be transmitted in families in a polygenic manner [30]. The set of genetic and epigenetic incidents transmitted at birth could explain the hereditary aspects, predispositions, and endometriosis-related changes in the endometrium. The heritability pattern of endometriosis was first proposed by Goodall in 1943, through his reference to five family history cases [31]. This genetic component of endometriosis can be dependent on the inherited allele differences in the enzymes involved in the development of the disease or may result from gene polymorphism [32]. By studies of siblings, certain alleles that occur more frequently than expected by chance, not consistent with random Mendelian segregation, have been identified. As a result, only the affected women and their ancestors are genotyped, and even smaller families can be recruited for genotyping [33]. In one study, the mothers and sisters of women with severe endometriosis had a seven-fold higher likelihood of developing endometriosis compared with primary female relatives of their partners [30].

In a clinical case-control study carried out on a British population sample of 64 women with endometriosis, 9.4% of the patients had a first-degree relative also suffering from the disease [34], whereas in another retrospective cohort study of 80 patients, endometriosis was detected in 5.9% of the patients’ first-degree relatives, which highlights the familial tendency of the disease [35]. Consistent with the above, another study reported that the patients’ sisters presented an 8.8% likelihood of endometriosis, with the relative risk amounting to 5.7%, in a total of 339 women with the disease [36]. The course of the disease in patients whose first-degree relatives were also affected is more severe than in women without such familial history. Such familial cases of endometriosis are characterized by an earlier onset of symptoms than sporadic cases [37]. Kennedy suggests that the risk for first-degree relatives of women with severe endometriosis is 6 to 9 times higher than that for relatives of unaffected women [38]. In another study, the assessment of 800 medical reports, including 400 surgical patients, suggested that the total risk for endometriosis of the patients’ first-degree relatives increased up to 10.2%, with the same percentage being only 0.7% for the controls [39]. A large study of 3096 twins based on the Australian Twin Registry demonstrated that the ratio of mono-zygotic to dizygotic twin pair correlations was over two-fold, suggesting that ~52% of the variance of the susceptibility to endometriosis may be attributable to additive genetic influences with negligible impact of environmental factors [40]. Two smaller twin studies reported that the concordance rate of endometriosis between monozygotic twins ranged from 75% to 87% [41,42]. Thus, twin and family studies have documented an increased relative risk of endometriosis, which has been investigated in numerous populations of European, Japanese, American, and Australian descent to confirm the previous associations and discover novel endometriosis risk loci. Several studies have genotyped the key SNPs from GWAS to replicate the association in different samples and ethnic groups of women.

In summary, it can be stated that endometriosis is a hereditary, multiple genetic disorder. Familial studies aim at defining inheritance trends. The accumulated data and observations emphasize the importance of assessing genetic variants in different ethnic and/or racial populations in an attempt to approach the genetic basis of endometriosis and the specific effects of various alleles in different populations [17,19,34,43,44,45,46].

### 2.2. Genetic Studies in Endometriosis

Genomics research allows to understand gene expression in the endometrium of patients with endometriosis and controls. Candidate genes have been identified to isolate the regions of genes that affect the risk of the disease. Additional linkage analysis studies have been conducted to map the specific genes along the entire genome that are likely to contain genetic polymorphisms related to the disease risk. Unfortunately, it is difficult for a linkage study to single out a specific gene or a gene variant [47]. To mitigate the difficulties in identification of the exact accountable gene or polymorphism, single-nucleotide polymorphism (SNP) association studies have been put into implementation [21,25]. The SNPs located close together are not necessarily independent of one another; related SNPs may be in linkage disequilibrium. As a result, identifying one SNP can help to pinpoint other SNPs related using association studies.

The Oxford Endometriosis Gene (OXEGENE) study, an international undertaking, aimed to identify the genetic loci associated with susceptibility to endometriosis using the linkage analysis technique [33]. For this purpose, the International Endogene Study, a conflation of the OXEGENE and the Genes Behind Endometriosis Australian project, examined over 1000 families in which there was a sibling suffering from endometriosis. The authors concluded that there are potential high-penetrance susceptibility loci on chromosomes 7p13–15, 10q26, and 20p13, and genes such as CYP2C19, INHBA, SFRP4, and HOXA10, which are probably responsible for the risk of development of endometriosis [45,48,49]. It was confirmed by later association analyses of the genes located in the identified regions with mutations responsible for the development of the disease [50,51].

However, no mutations strongly associated with the disease risk were identified in family-based or case control linkage and candidate gene studies for endometriosis [13,52,53].

### 2.3. Genome-Wide Association Studies in Endometriosis

Genome-wide association studies (GWASs) investigate new genomic regions associated with multifactorial disorders with the use of computational model techniques, aiming at the comparison of genotypes between patients and healthy individuals to identify endometriosis-related SNPs [54,55]. The patterns of SNP variation in the human genome have been characterized by the International HapMap Project; most of the common variations in the genome are tagged by SNPs [56]. High-throughput platforms make it possible to investigate up to a million SNPs in thousands of individuals in a single experiment [57]. Many GWAS-identified loci have been reported in women with endometriosis.

Non-coded areas, which cover about 95% of the human genome, are important sequences for the transcription and translational function of genes [14,58,59,60,61]. For example, if a polymorphism is in an intron or a promoter, it could affect the effective binding of a transcription factor, resulting in the expression of the corresponding protein at reduced levels. It has been suggested that the most common genetic factors contributing to the risk of endometriosis are located in the regulatory sequences of DNA and alter the regulation of gene transcription in the region [62].

Various candidate genes have been investigated to study the genetic background of endometriosis. The first significant evidence for genetic association was reported in two large GWAS studies. The first study, by Uno et al. [46], identified rs10965235 located in an intron of CDKN2BAS on chromosome 9p21 to be associated in a Japanese cohort, and in the second study conducted by Painter et al. [50], an association of SNP rs12700667 on chromosome 7p15.2 with advanced endometriosis was demonstrated in the UK and Australian cohort and was replicated in an independent cohort from the United States.

A meta-analysis of the subsequent studies based on GWASs extends this evidence and identifies a total of 11 independent SNPs associated with endometriosis [17]. These SNPs include: rs1519761 and rs6757804 on 2q23.3 identified in a US GWAS of European-ancestry women [63]; seven loci (rs7521902 near WNT4 on 1p36.12, rs13391619 in GREB1 on 2p25.1, rs4141819 on 2p14, rs7739264 near ID4 on 6p22.3, rs12700667 on 7p15.2, rs1537377 near CDKN2B-AS1 (independent of rs10965235) on 9p21.3 and rs10859871 near VEZT on 12q22) identified in a European ancestry GWAS [50] and from a meta-analysis of European and Japanese ancestry GWAS data [19]; and most recently rs17773813 near KDR on 4q12 and rs519664 in TTC39B on 9p22 in an Icelandic GWAS [64]. The suggested association of the IL1A gene locus on 2q13 has also been confirmed recently by identifying genome-wide significant association between rs6542095 and endometriosis [15]. The authors established that SNPs associated with endometriosis at the genome-wide significance level, of which all but one rs10965235 in CDKN2BAS on 9p21.3, identified in the Japanese GWAS [46] are polymorphic in populations of European ancestry.

In further studies, Rahmioglu et al. identified 27 genome-wide significant loci for endometriosis, 13 of which were novel associations. Among these 13 novel loci, seven were identified in the discovery meta-analysis and the remaining six reached genome-wide significance when the replication data were folded into the combined meta-analysis. The novel signals of association from the discovery meta-analysis with overall endometriosis were mapped to DNM3 on 1q24.3 (rs495590), near IGF2BP3 on 7p15.3 (rs62468795), near GDAP1 on 8q21.11 (rs10090060), in MLLT10 on 10p12.31 (rs1802669), in RNLS on 10q23.31 (rs796945), in RIN3 on 14q32.12 (rs7151531), and in SKAP1 on 17q21.32 (rs66683298). The replication analysis revealed six further novel loci: rs1894692 in an intergenic region between SLC19A2 and F5 on 1q24.2; rs2510770 in PDLIM5 on 4q22.3; rs13177597 near ATP6AP1L on 5q14.2; rs17727841 in IGF1 on 12q23.2; rs4923850 near BMF on 15q15.1; and rs76731691 in CEP112 on 17q24.1 [16]. These GWAS analyses revealed the genome-wide significant loci presented in Table 1.

ZNF366 has been proposed as a new contributor to the development of endometriosis, since four SNPs (rs227849, rs4703908, rs2479037, and rs966674) were found to be significantly associated with endometrioma risk. The genetic variant rs4703908 located near ZNF366 has been linked to an increased risk of endometrioma and deep infiltrating endometriosis [65].

In addition, Matalliotakis et al. [66] demonstrated that SNP rs11556218 is associated with the development of endometriosis, probably as a result of the aberrant expression of interleukin-16 (IL-16), which activates T-lymphocytes, leading to the secretion of several pro-inflammatory cytokines, resulting in the survival of the ectopic endometrial tissue in the peritoneal cavity.

### 2.4. Genes Associated with Endometriosis

A genome-wide gene-based analysis in Multi-marker Analysis of GenoMic Annotation (MAGMA) as implemented in functional mapping and annotation (FUMA) [67] allowed to identify 34 genes surviving the genome-wide significance threshold in endometriosis. Of these 34, 25 overlap with, or are located in the vicinity of, the genome-wide significant loci including 1p36.12 (WNT4, CDC42), 1q24.3 (DNM3), 2p25.1 (GREB1), 2q13 (IL1A), 2q35 (FN1), 6q25.1 (CCDC170, ZBTB2, RMND1, C6orf211, ESR1, SYNE1), 7p15.2 (RP1–170O19), 8q21.11 (GDAP1), 10p12.31 (MLLT10, DNAJC1), 10q23.31 (RNLS), 11p14.1 (ARL14EP), 12q22 (VEZT, FGD6), 12q23.2 (IGF1, NUP37, PARBP), 14q32.12 (RIN3), 15q15.1 (BMF), and 17q21.32 (SKAP1). The remaining nine genes were located in eight novel genomic regions including 3p25.3 (ATG7), 6p21.31 (HMGA1), 6q13 (CD109), 6q22.33 (RSPO3), 7q21.12 (ADAM22), 8q23.3 (TRPS1), 10p12.31 (SKIDA1), and 12q13.13 (HOXC6, RP11-834C11.12) [16]. The WNT4 and VEZT genes are the ones most consistently associated with endometriosis [43]. GWAS studies in women with endometriosis revealed numerous genes, including the genes associated with uterine development and stem cell function (WNT4), ovulatory function (ESR1, FSHB), and those regulating the activity of estrogen and estradiol biosynthesis (ESR1, GREB1, SYNE1, CYP2C19, CCDC170); most of these genes are also associated with ovarian cancer [68,69,70].

The genetic risk for endometriosis results from a large number of genetic variants, each of them exerting small effects. Despite identification of some genome-wide significant loci associated with endometriosis, no particular chromosome region occupied by the gene that can predict the risk of development of endometriosis in individual women in different ethnic groups has been identified to date [71]. It is noteworthy that the number of loci identified by GWASs is increasing as the proportion of the analyzed cases is limited to more severe stages of the disease, thus indicating that moderate to severe endometriosis cases have a greater genetic burden as compared to minimal or mild disease [72]. To date, these studies have not identified any therapeutically targetable molecules or gene products. Therefore, further detailed studies are needed in each region to identify the causal SNPs and target genes.

## 3. Steroidogenic Pathway

Endometriosis is an estrogen-dependent disease associated with suppression of progesterone receptors; therefore, the search for and identification of the regulators of their receptors has been of key importance for research on progression of the disease [1]. Estrogen initiates proliferation of the endometrial tissue and supports the growth of the endometrial glands before ovulation, preparing the endometrium for the action of progesterone. Estrogen and progesterone act by binding to their intracellular receptors, the estrogen receptor (ER) and progesterone receptor (PR), members of the steroid/nuclear receptor (SR) superfamily [73]. These receptors are currently extensively investigated steroid receptors involved in the pathophysiology of endometriosis [74]. It has been demonstrated that there are several abnormalities in the intracavitary endometrium and ectopic endometriotic tissue underlying endometriosis progression: inflammation activated by excess estrogen biosynthesis, defective differentiation due to progesterone resistance, dysregulated differentiation of endometrial mesenchymal cells, and abnormal epigenetic marks. Among these, hypoxia and inflammation play an important role in the regulation of the steroidogenic pathway in the development of endometriosis [75,76]. In endometriosis, progesterone and estrogen signaling are disrupted, commonly resulting in progesterone resistance and estrogen dominance [74]. Additionally, estrogens are the activators of prostaglandin (PGE2) synthesis. Inhibition of PGE2 biosynthesis impedes growth of endometriosis and endometriosis-related enhanced inflammatory condition, chronic pelvic pain, and infertility in women [74,77]. Endometriotic stromal cells are capable of synthesizing estradiol from cholesterol via the steroidogenic pathway, whereas normal endometrial stromal cells do not produce steroid hormones. Endometriotic lesions aberrantly overexpress the entire repertoire of steroidogenic enzymes including StAR, CYP11A1, CYP17A1, and CYP19A1 [74]. They display a wide range of abnormal expression of nuclear receptors, which compete to regulate the steroid-synthesizing genes. There are two types of estrogen receptors: estrogen receptor type 1 (ESR1) and estrogen receptor type 2 (ESR2). Both receptors are biologically active and demonstrate different tissue specificity and different gene activation patterns. The ESR1 gene is localized in the q24-q27 portion of chromosome 6, and the ESR2 gene in the q23.2 stria of chromosome 14. The ratio of expression of ESR1 to ESR2 receptors may alter a cell’s receptivity to estrogen. It has been observed that in endometriotic stromal cells ESR2 levels are 142-fold higher and ESR1 levels are 9-fold lower compared with normal endometrium [78]. Han et al. [79] postulated that increased progesterone resistance downregulates ESR1 expression, which means that early endometriotic tissue may carry more ESR1 than older tissue. Endometriotic stromal cells are also deficient in the progesterone receptor (PGR) gene, which leads to progesterone resistance and defective retinoid synthesis [80]. Progesterone via the PR increases the formation of retinoic acid (RA) in endometrial stromal cells, which in turn induces 17β-hydroxysteroid dehydrogenase gene type 2 (HSD17B2) expression in the adjacent endometrial epithelial cells [81,82]. In contrast, endometriotic stromal cells demonstrating progesterone resistance do not produce RA [83], which leads to a loss of paracrine signaling to induce HSD17B2 expression in the epithelial cells, and consequently a failure to inactivate estradiol in endometriosis [83,84].

As endometriosis is recognized as a steroid-dependent disorder, numerous researchers have investigated the genes of steroid biosynthesis and signaling [17,47,85]. However, the critical genes that confer this steroidogenic transformation have not been defined to date [74,76]. The physiological and pathologic activities of sex steroids are mediated through the estrogen receptor ESR1/ESR2 genes and the PGR gene. The genes for progesterone and estrogen receptors, including ER, PR, HSD17B1, CYP17, and CYP19A1, have been found to have an association with endometriosis. The genes for estrogen receptors are found at different chromosomal locations [86]. The CYP17 gene is located on chromosome 10q24.3 and encodes the cytochrome P450c17a enzyme, involved in the biosynthesis pathway of sex steroids through 17ahydroxylase and 17,20-lyase activities.

Sapkota et al. [17] identified several independent signals in the region that includes ESR1 encoding estrogen receptor 1 on chromosome 6p25.1. In primary meta-analysis, they identified two SNPs at the locus rs71575922 in SYNE1 and rs1971256 in CCDC170, located in endometriosis up- and downstream of ESR1, respectively. In addition, Dunning et al. further identified two independent associations at that locus, including rs17803970 in SYNE1 and rs2206949 in ESR1 [59].

Further support for association at the 2p25.1 locus, containing an estrogen-regulated gene, GREB1 for secondary association with the risk of endometriosis was provided by Sapkota et al. [17]. Regulation of GREB1 transcription by ESR1 is mediated through three estrogen response elements located 20 kb upstream of the gene [87]. Additionally, GREB1 is an essential component of the estrogen receptor transcription complex, and despite the fact that the impact of the individual risk SNPs is small, the research results suggest that risk variants acting on several genes in the same pathway cause an increase in sensitivity to estrogen, thus increasing the risk of development of endometriosis [88].

## 4. Steroidogenic Factor-1

Identifying the regulators of steroidogenic pathway plays an essential role in the development and maintenance of endometriosis [24,73,74,89].

There are two known orphan nuclear receptors which modulate steroidogenesis and focus on the impact on reproductive processes: steroidogenic factor-1 (NR5A1, also known as SF-1) and liver receptor homolog-1 (NR5A2, also known as LRH-1). They bind to the same DNA sequences and display differing and often non-overlapping effects, in particular, on reproductive target tissues [90]. SF-1 is expressed primarily in steroidogenic tissues, while LRH-1 is expressed in tissues of endodermal origin and the gonads. The human SF-1 gene is located on chromosome 9 stria q33 and consists of seven exons. Exon 1 encodes 5’-UTR (untranslated region), while exons 2 and 3 encode the DNA-binding domain (DBD) [76,90].

SF-1 is a key transcriptional factor regulating the expression of many genes involved in estrogen biosynthesis and steroidogenesis. SF-1 has an important role in a variety of biological processes including cell proliferation, apoptosis, and angiogenesis [76,90].

In endometriosis SF-1, besides contributing to local steroidogenesis and growth of ectopic endometrial tissue, it disrupts multiple signaling pathways, triggers a physiological inflammatory response and alters endometrial immune homeostasis, inhibits the ability of the endometrium to undergo decidualization resulting in infertility, and promotes the abnormal uterine gland morphogenesis [91].

Previous studies have reported aberrant SF-1 expression in endometriotic tissues and stromal cells compared with eutopic endometrial tissues and stromal cells [78,92,93]. There is considerable evidence that both SF-1 mRNA and SF-1 protein levels in endometriotic cells are significantly higher than those in eutopic endometrial cells [78,89]. Such increased SF-1 expression or activity in endometriotic tissue could result in increased activity of steroid acute regulatory protein (StAR) and aromatase (CYP19A1), resulting in increased local estrogen biosynthesis, a key pathological feature of endometriosis [78,89,93]. In turn, the absence of SF-1 in endometrial cells underlies the lack of responsiveness of steroidogenic genes to PGE2, which, acting through SF-1, stimulates the expression of the genes for steroidogenic enzymes, mainly StAR and CYP19A1 [74,76]. Thus, most probably, a vicious circle mechanism arises in the etiopathogenesis of endometriosis, in which locally produced estrogen intensifies inflammation, whose mediators in turn stimulate estrogen steroidogenesis. Epigenetic alterations of the chromatin landscape of endometrial tissue of some women are hypothesized to result in molecular abnormalities that subsequently functionally disrupt normal responsiveness to steroidogenesis [73,74].

The mechanisms that regulate SF-1 expression in endometriosis are not fully understood. The SF-1 expression is under epigenetic control that permits binding of the activator complexes to the SF-1 promoter [78,93]. The abnormal expression of SF-1 in endometriosis may be caused by epigenetic modifications determined primarily by the methylation of its promoter [78]. The CpG (cytosine that precedes guanosine) island which flanks the SF-1 promoter and exon 1 region has been noted to be hypomethylated in endometriotic cells compared with the normal endometrium [78]. In the later studies, the same authors reported that hypermethylation of the CpG island that spans from exon 2 to intron 3 of the SF-1 gene activated mRNA expression in endometriotic cells [94]. These observations allowed to hypothesize that the hypermethylation of this particular region of the gene, distant to the promotor, included a silencer which, when hypermethylated, suppresses its silencer function, giving rise to increased SF-1 expression.

In addition to interaction with methylating enzymes, interaction with demethylating enzymes, such as DNA methyltransferases (DNMT3B), may be the cause of abnormal expression of SF-1 in endometriosis [24,73]. Aberrant demethylation of the SF-1 promoter in endometriosis results in the upregulated expression of SF-1 [78,93]. Acetylation of histone H3 and H4 could also be the cause of increased expression observed in pathologic cells [73]. In addition, miRNAs mediate SF-1 expression. It has been demonstrated recently that miR-370-3p functions as a negative regulator of SF-1 and cell proliferation in endometriotic cells [95]. In addition to the miRNAs mentioned above, reduced levels of miR-23a and miR-23b expression were confirmed in ectopic and eutopic endometrium, from patients with endometriosis compared with normal endometrium from negative and this reduction was associated with elevated transcript levels of SF-1, StAR, and CYP19A1 [96]. Our findings provide further insight into the molecular mechanisms underlying local estrogen production in endometriosis.

Considering the pivotal role of transcription factor in endometriosis, investigating the regulators of SF-1 may open a novel strategy for treatment of the disease.

## 5. Epigenetic Processes in Endometriosis

The cells of the human body, despite having the same genome, show a huge variety of forms and functions, which allows them to form tissues and organs different phenotypically and functionally. Epigenetic phenomena, which precisely regulate which gene is to be expressed at a given moment, are responsible for such tissue-specific gene expression profiles. Epigenetic research has been focused on the changes in gene function acquired following mitosis and/or meiosis processes that cannot be explained and justified by changes in DNA sequence.

Epigenetic processes seem to be important in the pathomechanism of complex human diseases. In the search for explanations of endometriosis pathomechanisms, especially the development of its complications (pain and infertility), attention was drawn to the role of epigenetic inheritance associated with epigenetic modifications. Considerable impact is exerted by environmental factors, affecting the epigenome which leads to the onset of the disease. In this regard, the epigenome as well as the hormonal and immune status influence each other, contributing to the development of endometriosis [74]. In connection with the above, a new theory regarding the pathogenesis of endometriosis has been proposed. According to it, the set of genetic and epigenetic incidents transmitted at birth could explain the hereditary predispositions to endometriosis [97].

The basic epigenetic mechanisms include DNA methylation and histone modification (methylation or acetylation of specific histones in chromatin), as well as the involvement of non-coding RNA (ncRNA), such as miRNA or siRNA (Figure 2) [73,74]. Epigenomic mechanisms affect all genes and intergenic regions in DNA, which are packaged together with proteins into chromatin. The alteration of chromatin conformation constitutes the basis of epigenetic regulation [73]. Epigenetic mechanisms can affect transcript stability, DNA folding, nucleosome positioning, chromatin compaction, nuclear organization, and finally the results of determining if a gene can be expressed or silenced [73,98]. Epigenetic regulation is a multistage process and can be modulated at each stage. Accumulating evidence suggests that various epigenetic aberrations may play an essential role in elucidation of the pathogenesis of endometriosis [73]. Interestingly, these epigenetic aberrations, due to their dynamic and reversible nature, may have potential implications for diagnosis, prognosis, and therapy of the disease.

### 5.1. DNA Methylation in Endometriosis

Methylation of DNA is the best known change leading to gene inactivation in humans among epigenetic modifications. DNA methylation is a process in which methyl groups are added to CpG (cytosine that precedes guanosine) islands located in the promoter regions of genes in order to silence gene expression [24,99,100]. Over 40,000 CpG islands, recognized as being differentially methylated in endometriosis have been identified [26,99].

The family of DNA methyltransferases (DNMTs) (deoxyribonucleic acid methyltransferases) catalyzed by a group of DNA methyltransferases (DNMTs), consisting of DNMT1, DNMT3A, and DNMT3B, is responsible for DNA methylation. The role of DNMT1 is to perpetuate the methylation pattern (methylation maintenance) after DNA replication [101], whereas DNMT3A and DNMT3B are mainly involved in methylation of new sites, known as de novo methylation and their highest expression occurs during embryogenesis [102,103,104].

DNMTs have differential expression patterns in endometriotic tissue compared with normal endometrium, as evidenced by inconsistent results of a few studies that were conducted in a group of women with endometriosis. The overall DNMT expression decreases in endometriotic cells as the endometrium passes to the secretory phase [100]. The DNMT1, DNMT3A, and DNMT3B expression levels were reported to be elevated in ectopic endometria compared to normal controls or in the eutopic endometrium of women with endometriosis. In addition, the expression levels of DNMT1, DNMT3A, and DNMT3B were demonstrated to correlate positively with each other [105].

The described upregulated expression of DNMTs in the endometriotic tissue, which leads to hypermethylation (the gene that will be silenced is methylated), has been clearly observed in eutopic endometrium in infertile women with endometriosis only for the DNMT3A transcript and not for DNMT1 and DNMT3B [106]. Conversely, for hypomethylation (the gene that will not be silenced is not methylated), lower expression of DNMT1 has been demonstrated in eutopic and ectopic endometria of endometriosis patients compared with that in control endometria, and the level of DNMT3B was significantly lower in ectopic endometria compared with eutopic and control endometria [107,108]. Wang et al. clarified previous findings and showed that the expression levels of all three DNMTs were significantly lower in endometriotic lesions and eutopic endometria compared with control endometrial of Northern Chinese women [109].

Thus, it can be assumed that the maintenance of global CpG methylation in endometriosis may depend on expression of a combination of DNMTs. These data provide information valuable for further understanding the role of DNA methylation in the pathogenesis and progression of endometriosis [110]. It has been suggested that aberrant DNA methylation in endometriotic lesions represents the potential mechanism that may be linked to some genetic factors, immune and inflammatory responses, defective estrogen metabolism, and environmental factors [111].

Hypoxia, inflammation, and steroidogenic pathway contribute to the development of endometriosis [24,73,108], distinctly modulating the expression of DNMTs. They can act together to cause aberrant DNA methylation patterns [73,112,113]. Although both hypomethylated and hypermethylated DNA for specific genes have been reported in endometriotic epithelial and stromal cells, a recent study revealed global methylation decreases in ectopic stromal cells, which is mainly caused by hypoxia-mediated DNMT1 downregulation [73,113]. In contrast to the suppressive effect of hypoxia on DNMT1, a selective blockage pathway of the prostaglandin E2 (PGE2) receptors EP2 and EP4 has no effect on the level of DNMT1 but suppresses DNMT3A expression [112]. The maintenance of DNMT3A level by the inflammation pathway has been implied, since inhibition of PGE2-EP2/EP4 biosynthesis inhibits growth, invasion, migration, adhesion, and survival of endometriotic epithelial and stromal cells by upregulating proteins associated with these pathways, thus impeding growth of endometriosis (Figure 3) [112].

The expression of DNTMs changes in normal endometrium in response to steroid hormones [73,74,89]. As the genes coding for steroid hormones are involved in the pathogenesis of endometriosis, it is possible that DNMT1 expression in the endometrial epithelium is more sensitive to steroid hormones, and as van Kaam et al. point out, the epithelial response to steroids would be affected by the presence or absence of stromal cells [107].

DNMT3B has been demonstrated to bind to the promoter regions of the key steroidogenic genes, SF1 and ESR1, in endometrial vs. endometriotic stromal cells [114]. DNMT3B has been postulated to be the cause of abnormal expression of SF-1 [100]. Thus, differentiated DNMT3B expression and binding to critical gene promoters in endometriotic stromal cells may contribute to aberrant DNA methylation that misdirects gene expression in endometriosis and contributes to altered response of these cells to steroid hormones [114]. Moreover, hypomethylation of the CpG island at the promoter region of the ESR2 gene leads to high levels of expression in endometriotic stromal cells, and hypermethylation silences the ESR2 gene in endometrial stromal cells. In contrast, the ESR1 promoter is unmethylated in eutopic endometrium and heavily methylated in endometriosis [99], leading to lower ESR1 receptor levels in endometriotic vs. endometrial stromal cells [78].

Research has indicated that the intrauterine environment altered due to endometriosis demonstrates abnormalities in DNA methylation, which causes a change in gene expression and progesterone resistance both in eutopic endometrial tissue and in the developing lesions [115,116]. Silencing of progesterone and aromatase genes through promoter hypermethylation may contribute to the development of endometriosis. In turn, targeting EP2 and EP4 receptors may be effective as a long term non-steroidal therapy for treatment of active endometriotic lesions.

### 5.2. Epigenetic Histone Modifications

Post-translational histone modifications can lead to changes in chromatin structure and conformations, resulting in loosening of the gene structure and transcription of genes or, on the contrary, in condensation and thereby inhibition of gene expression.

Histone modifications and DNA methylation are interrelated in regulation of chromatin remodeling and gene expression [73,117]. It has been observed that transcriptional gene activation is related to hypomethylation, whereas the transcriptionally non-active genes are hypermethylated. Hypermethylation has been noted to occur at the ends of chromosomes in endometriotic stromal cells, suggesting that methylation changes are not random [117]. Histone methylation is a prerequisite for DNA methylation [112].

Histone modifications include methylation, acetylation, phosphorylation, ubiquitination, or conjugation with small ubiquitin-like modifier (SUMO) molecules, so-called sumoylation (Table 2) [118].

Currently, the best-known modifications of histones include acetylation and methylation of histone proteins. These processes occur within the arginine or histone lysine residues. In the acetylation process, the chromatin-modifying enzymes are histone acetyltransferases (HATs) and histone deacetylases (HDACs) and in the process of methylation-histone methyltransferase (HMTs) and antagonistic histone demethylases (HDMTs) [119]. Acetylase and deacetylase maintain the systemic homeostasis by stimulating cell growth, differentiating myotubules, proliferation of adipocytes, or regulation of myofilament contractility [120].

As far as the epigenetic regulation by histone acetylation is concerned, the balance between the HDACs and HATs activity regulates the gene transcription. Gene expression is promoted by the acetylation of lysine residue by HATs, whereas it is inhibited through the removal of the acetyl group by HDACs [121].

Arginine methylation activates expression only, while lysine methylation can cause both activation and suppression of transcription. Methylation of histone 3 (H3) lysines is most important for activation, whereas methylation of both H3 and histone 4 (H4) is important for suppression. The histone methylation process does not directly affect the structural changes of chromatin but creates binding places for other proteins that may affect its condensation [122].

Not only acetylation, but also phosphorylation of histones lead to increased gene expression by loosening the structure of chromatin. It is the effect of histone kinases and includes modifications to the rest of serine and threonine residues. This process affects the degree of condensation of chromatin during mitosis [123]. Ubiquitination activates the transcription, and sumoylation probably silences it by influencing methylation and deacetylation of histones [124]. In addition, the sumoylation process can affect DNA repair, modification of chromatin structure, cellular proliferation, or apoptosis [125].

Modifications of histone proteins are more complex than DNA methylation as they are associated with a larger number of post-translational modifications of histones, which are dependent not only on the type of modification, but also on the locus where such modification occurs on the histone protein and on binding of different numbers and different additional molecules or function groups. This gives a huge number of possible “combinations” affecting the chromatin structure and gene expression, which leads to a variety of effects.

Epigenetic modifications play an important role during the pathogenesis of endometriosis [73,74,110]. Variation in gene expression in human endometrium is strongly influenced by the stage of the menstrual cycle. The global methylation level of eutopic endometrium was reported to be higher in the proliferative phase as compared to the secretory phase of the menstrual cycle [126].

It is known that different genes are involved in the etiopathogenesis of endometriosis by methylation of their promoters and subsequent downregulation. Transcriptional gene activation is supposed to be related to hypomethylation and the transcriptionally non-active sequences are thought to be hypermethylated. Promoter hypermethylation may contribute to the understanding of epigenetic regulation in endometriosis. DNA hypermethylation in endometriosis affects the expression of several key genes. They include, among others, the genes encoding insulin-like growth factor binding protein 1 (IGFBP1), SF-1, and CYP19A1, including PR-B and HOXA10 [99].

Histone modifications by acetylation seem to be involved in endometrial function; histone acetylation levels were reported as globally increased in the early proliferative phase and gradually reduced in the late proliferative phase until ovulation [127]. Global histone acetylation profiles have shown that certain histones, commonly H3 and H4 are hypoacetylated in endometriotic stromal cells compared with normal endometrium [128,129]. Monteiro and colleagues postulated the H3 and H4 histones in the promoter region of ESR1 to be hypoacetylated [129]. As suggested by the research, the etiology of endometriosis may be partially explained by epigenetic regulation of gene expression due to dysregulation in the expression of HADCs. Increased activity of HDAC in endometriotic cells has been shown to leave promoter regions hypoacetylated, which leads to cell cycle induction and proliferation [117]. Histone deacetylase 1 (HDAC1), HDAC3, and two histone acetylases were reported to be constitutively expressed in the endometrium during the menstrual cycle, with a reduced HDAC1 level demonstrated in the secretory phase [130]. The expression levels of HDAC1 and HDAC2 were significantly downregulated by estradiol and progesterone in endometrial epithelial cells, whereas the expression levels of HDAC2 were upregulated by estradiol and downregulated by estradiol + progesterone in endometrial stromal cells [73,131]. HATs, such as steroid receptor coactivator (SRC-1), p300, and cyclic adenosine monophosphate response element binding protein (CBP), are required for the actively expressed SF-1 and StAR genes in endometriotic cells [73]. The above results suggest that histone modifications may play a role in the control of decidualization through the regulation of the function of ERs and PGE2-EP2/EP4-induced 17 β-estradiol synthesis [73,132,133,134]. It is noteworthy that the effects of PGE2-EP2/EP4 inhibition may be due to epigenetic mechanisms such as DNA methylation and histone modification (Figure 3).

Several genes that were hypoacetylated in endometriosis have been identified. They include ESR1, p16 (INK4a, CDKN2A), p21 (Waf1/Cip1, CDKN1A), p27 (Kip1, CDKN1B), death receptor 6 (DR6), checkpoint kinase 2 (CHEK2), homeobox A10 (HOXA10), E-cadherin (CDH1), and CCAAT/enhancer-binding protein alpha (CEBPA) [121,135]. The existence of a “histone code”, the specific histone modification as histone tails that have regulatory effects in a small number of target genes has been suggested. Deciphering that code may help in understanding the genes causing gene deregulation in endometriosis [121].

Endometriotic lesions, as well as eutopic endometrium with endometriosis were reported in different populations to be hypoacetylated as compared to the eutopic endometrium of the control, with consequent gene silencing [129]. This observation was confirmed by the reported higher expression of HDAC1 and HDAC2 genes and lower levels of SIRT1 in the endometriotic lesions of affected women in comparison with normal endometrium. Moreover, in ectopic implants, the loss of HDAC expression modulation by estrogen and progesterone was reported [128,129,136]. Studies concerning histone modification are still scarce and heterogeneous in results, method, and study design. Histone deacetylation, like promoter methylation, generally results in gene silencing thus HDACs act as transcriptional repressors in endometriosis [131]. The therapeutic and prognostic implications are based on epigenetic modifications which are reversible. Therefore, the enzymes involved in epigenetic mechanisms could become the targets of pharmacological interventions. The fact that modulating histone acetylation might ameliorate endometriosis is particularly noteworthy [73]. The application of HDAC inhibitors (HDACI) to cause histone hyperacetylation inhibits mitosis and DNA damage responses. HDACIs are able to move the cells from the silenced chromatin state, towards activation and differentiation, thus limiting proliferation [65,137]. HDACIs added to endometriotic cells have been demonstrated to promote acetylation of H3 and H4 in the promoter region of cyclin-dependent kinase (CDK) genes leading to the cessation of cell proliferation by suppressing the genes whose promoters these histones are located upon [121]. Notably, HDACIs were able to determine the morphological transformation and differentiation in the endometrium, which suggest that their application as therapeutic agents for endometriosis should take into consideration the localization of lesions and the specific profile of expression of HDAC isoforms [65,73,131,138]. That is why HDACIs appear to be a promising target to improve the treatment of endometriosis in the future.

### 5.3. Role of Non-Coding RNA in the Detection of Endometriosis

In contrast to the epigenetic mechanisms mentioned above, non-coding RNAs (ncRNAs) regulate gene expression at the post-transcriptional level. It is noteworthy that both epigenetic changes of DNA methylation nature and changes within histone proteins affect the expression of genetic information by affecting ncRNA activity. To date, a dozen or so types of ncRNAs have been identified, including microRNAs (miRNAs), small interfering RNAs (siRNAs), and piwi-interacting RNAs (piRNAs). PiRNAs, like siRNAs, take part in maintaining genome integrity by silencing transposition elements [139,140]. According to their size, three main ncRNA classes were distinguished: small ncRNA with a length of less than 200 nt; long ncRNA, more than 200 nt long and transcripts of intragenic areas; and very long ncRNAs containing hundreds of thousands of bases, involved in the regulation of intergenic sequences [140]. The most numerous class, however, is the long ncRNAs (lncRNAs). LncRNA expression changes in response to stress and other environmental signals. Their role is to participate in the regulation of intercellular transport, the recruitment of transcription factors, RNA processing, and formation of ribonucleoprotein complexes [140].

MiRNAs are small single-stranded non-coding RNAs, containing on average 22 nucleotides. They are capable of modifying gene expression and play important regulatory roles by targeting matrix RNAs (mRNAs) for cleavage or transcription/translation repression [141].

As demonstrated by GWASs, aberrant miRNA expression profiles play critical roles during the development of endometriosis through modulating cell cycle progression, apoptosis, proliferation, steroidogenic pathway, hormone signaling, inflammation, and response to hypoxia [73]. Thus, these molecules play an important role in maintaining body homeostasis.

Numerous studies have demonstrated that miRNA expression is altered in both ectopic and eutopic endometrium tissues in women with endometriosis as compared to healthy women. However, the studies vary in which miRNAs they address and what directions their expression patterns change [142,143,144,145,146,147]. Over 50 different miRNAs have been shown in various studies to be differentially expressed in endometriotic cells. The most studied miRNAs associated with endometriosis include the miR-200 family, miR-20a, miR-143, 145, miR199a, and let-7 [10,148]. Assessments of specific miRNAs expression by microarray analysis in paired ectopic and eutopic endometrial tissues of women with endometriosis have been reported and many upregulated (miR-1, miR-29c, miR-99a, miR-99b, miR-100, miR-125a, miR-125b, miR-126, miR-143, miR-145, miR-150, miR-194, miR-223, miR-342, miR-365, miR-370, miR-375, and miR-451a) and downregulated (miR-20a, miR-34, miR-141, miR-142-3p, miR-183, miR-196b, miR-200a, miR-200b, miR-424, miR-3613, and let-7b) miRNAs [20,95,149,150,151,152] have been identified. As demonstrated recently, the molecular constituents of exosomes, especially exosomal miRNAs, may be novel promising biomarkers for the diagnosis of endometriosis. Exosomal miRNAs such as miR-22-3p and miR-320a, which were significantly elevated in the serum exosomes of women with endometriosis, have been identified [153]. Most of these identified miRNAs target genes that are known to be differentially expressed in eutopic vs. ectopic endometrium. Approximately 30% of all human genes are probably regulated by miRNAs [154,155]. MiRNA genes have different locations, located in introns and/or in structural gene exons or intergenic areas. They can occur individually or in clusters, having common regulatory sequences [60,61]. Moreover, miRNAs have been reported to be both targets and regulators of other epigenetic mechanisms such as methylation and acetylation, and they resulted in involvement in the hypoxia and inflammation signaling pathways. The list of genes, the transcription of which is affected by miRNAs, includes the coding ones, namely HMTs, HDACs, and proteins with chromodomains binding the methylated lysine and arginine residues [73,152,154] as well as the identified miRNA target genes including those involved in hormone metabolism such as ESR1, ESR2, PR, and aromatase; modulators of the inflammatory response such as IL-6, IL-8, tumor growth factor (TGF)-β, and cyclo-oxygenase type 2 (COX-2); and the induction of apoptosis and angiogenesis such as vascular endothelial growth factor (VEGF), Bcl-2, and cyclin-D [73,74]. It has been demonstrated that endometriotic tissues have significantly increased levels of mRNA encoding through steroidogenic genes, including StAR, CYP17, and CYP19A1, which may contribute to estrogen levels [156]. A mismatch between the expression of transcriptomes and proteins associated with endometriosis has been demonstrated. The existing evidence supports the ability of many miRNAs to interact with transcription factors forming a network for gene regulation that yields both negative and positive feedback loops [74,157]. The aberrant expression patterns of some specific miRNAs and genes identified in endometriosis confirm the hypothesis that they are involved in the pathogenesis of this disease [74]. Owing to the development of high-throughput sequencing and other biomolecular technologies, comprehensive studies have been conducted at multiple biological levels by using the “omics” platform, which allows a better understanding of genome-wide epigenetics in endometriosis [158].

The discovery of epigenetic phenomena as the determinants of gene expression has created a new chapter in the treatment of the disease. Knowledge of the processes modulating gene expression and, consequently, indirectly affecting protein responses, as well as the use of miRNAs as markers, have given rise to the possibility of developing targeted therapeutic strategies interfering with the pathological processes at the very source. Elucidating the mechanisms and pathways involved in the pathogenesis of endometriosis can enable the development of more specific means of prevention and therapy of the disease. Novel therapies based on miRNAs are under development and seem promising for endometriosis.

## 6. Conclusions

The cause of endometriosis remains unknown to date, and its complex etiopathogenesis has been only partially elucidated. In addition to familial predisposition and genetic causes of endometriosis, multiple theories have been postulated, including epigenetic influences. Conspicuous progress in this area has been achieved, mainly due to the identification of new candidate genes and numerous SNPs closely associated with endometriosis. The genes known to be associated with abnormal modulation of cell cycle progression, apoptosis, adhesion, angiogenesis, proliferation, immune and inflammatory processes, response to hypoxia, steroidogenic pathway, and hormone signaling are involved in the pathogenesis of endometriosis. Familial studies, linkage analyses, genetic association studies, and GWASs have contributed to partial elucidation of the pathophysiology of the disease. They have resulted in identification of the loci significant for the whole genome, posing the risk of the disease, which alter a woman’s risk of developing the disorder, and have provided new information concerning the potential pathways leading to endometriosis. However, the accumulated research results are conflicting and indicate the existence of a number of differences in associations of the disease with the frequency of genetic polymorphisms in women representing different world populations. Therefore, the analysis of epigenetic changes seems highly desirable due to their involvement in the development of endometriosis observed with increasing frequency. Understanding the modification mechanisms, including DNA methylation, reorganization in chromatin structure related to the functioning of histone proteins, or changes in non-coding RNA expression, provide the possibility of developing more effective therapies. Thanks to modern technologies for detecting epigenetic changes, there is a chance to discover new biomarkers that could contribute significantly to the detection of endometriosis already at an early stage of its development. Therefore, research into broadly understood epigenetic changes is highly desirable, because the obtained results can be used in the future to construct a new effective strategy for the treatment of endometriosis.

## Figures and Tables

**Figure 1 jcm-09-01309-f001:**
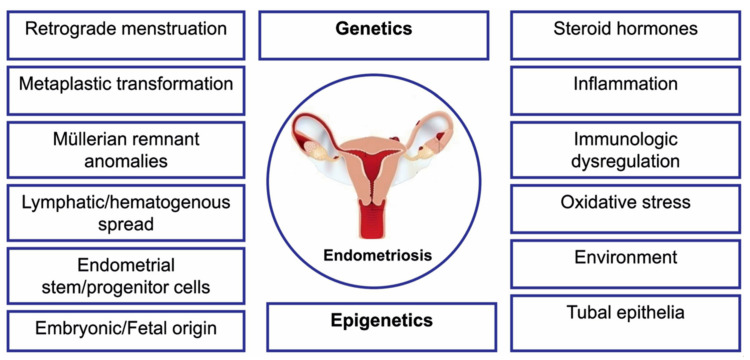
Summary model for the pathogenesis of endometriosis.

**Figure 2 jcm-09-01309-f002:**
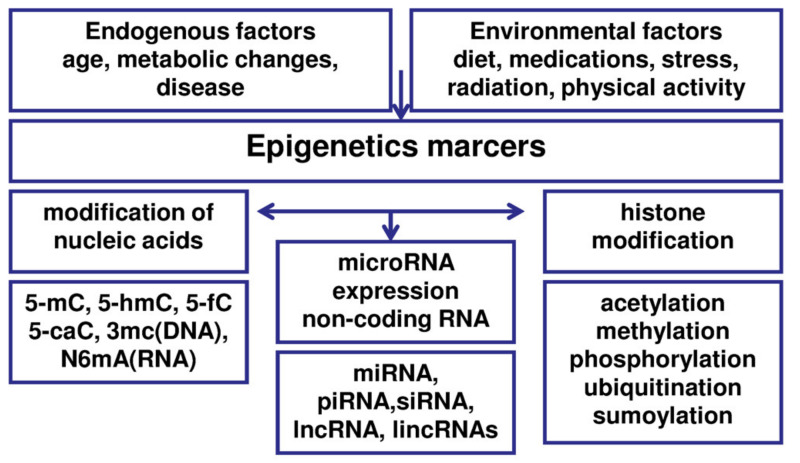
Epigenetic regulation of gene expression in endometriosis.

**Figure 3 jcm-09-01309-f003:**
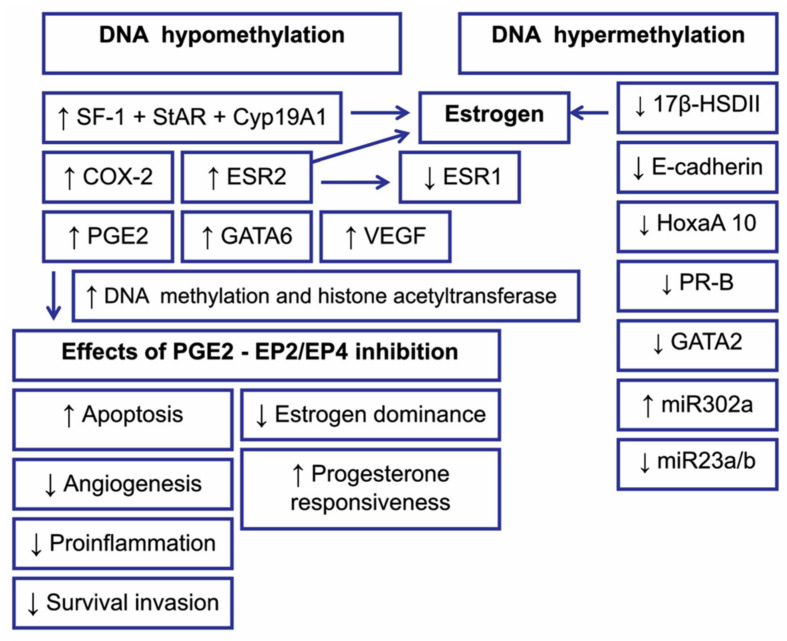
DNA hypo- and hypermethylation in endometriosis. Effects of PGE2-EP2/EP4 inhibition may be due to epigenetic mechanisms such as DNA methylation and histone modification.

**Table 1 jcm-09-01309-t001:** Review of the selected genetic polymorphisms for the significant loci in the entire genome associated with the development of endometriosis.

Chr	SNP	Associated Gene/Cytoband	References
1 1 1 1 1 2 2 2 2 2 2 2 2 4 4 4 4 5 6 6 6 6 6 6 6 7 7 7 7 8 9 9 9 9 9 9 10 10 11 12 12 12 12 14 15 17 17	rs12037376 rs7521902 rs16826658 rs1894692 rs495590 rs11674184 rs77294520 rs13394619 rs4141819 rs654324 rs10167914 rs1250247 rs1250241 rs10012589 rs1903068 rs17773813 rs2510770 rs13177597 rs6938760 rs760794 rs7759516 rs1971256 rs2206949 rs17803970 rs71575922 rs12700667 rs62468795 rs55909142 rs74491657 rs10090060 rs9987548 rs1537377 rs10757272 rs1448792 rs10965235 rs519664 rs1802669 rs796945 rs74485684 rs12320196 rs4762326 rs10859871 rs17727841 rs7151531 rs4923850 rs66683298 rs76731691	WNT4/1p36.12 WNT4/1p36.12 WNT4/1p36 SLC19A2/1q24.2 DNM3/1q24.3 GREB1/2p25.1 GREB1/2p25.1 GREB1/2p25.1 ETAA1/2p14 ETAA1/2p14 IL1AI/2q13 FN1/2q35 FN1/2q35 KDR/4q12 KDR/4q12 VEGFR2/4q12 PDLIM5/4q22.3 ATP6AP1L/5q14.2 ID4/6p22.3 ID4/6p22.3 CCDC170/6q25.1 CCDC170/6q25.1 ESR1/6q25.1 SYNE1/6q25.1 SYNE1/6q25.1 7p15.2/7p15.2 IGF2BP3/7p15.3 7p12.3/7p12.3 7p12.3/7p12.3 GDAP1/8q21.11 CDKN2-BAS1/9p21.3 CDKN2-BAS1/9p21.3 CDKN2-BAS1/9p21.3 CDKN2-BAS1/9p21.3 CDKN2-BAS1/9p21 TTC39B/9p22 MLLT10/10p12.31 RNLS/10q23.31 FSHB/11p14.1 VEZT/12q22 VEZT/12q22 VEZT/12q22 IGF1/12q23.2 RIN3/14q32.12 BMF/15q15.1 SKAP1/17q21.32 CEP112/17q24.1	[16,17] [19] [46] [16] [16] [16,17] [17] [19] [15,16,19] [17] [16,17] [16] [17] [16] [17] [64] [16] [16] [16] [17] [16] [17] [17] [17] [16,17] [15,16,17,19,50] [16] [16] [17] [16] [16] [15,17,19] [17] [17] [19,46] [64] [16] [16] [16,17] [16] [17] [19] [16] [16] [16] [16] [16]

Chr, chromosome; SNP, single-nucleotide polymorphism.

**Table 2 jcm-09-01309-t002:** Effects of histone modifications.

Modification	Global Effect of Modification
Acetylation	Activation of Transcription Silencing of Telomeres DNA Repair
Methylation	Inactivation of transcription
Phosphorylation	DNA repair Mitosis
Ubiquitination	Activation of transcription
Sumoylation	Silencing of transcription

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
