# Peer review of "Genetic, Epigenetic, and Steroidogenic Modulation Mechanisms in Endometriosis"

_jcm, 2020, doi:10.3390/jcm9051309_

Round 1

Reviewer 1 Report

Zubrzycka et al from Poland have written this review of endometriosis mainly in the aspects of genetic and epigenetic regulatory mechanisms.  Overall, the review is well written and the knowledge in the field is reasonably updated.  I don’t have any major concerns about the contents and the way of presentation of the review.  However, I do have the following minor points for authors’ consideration while revising the manuscript. 

  1. Title: The title of this review basically covers genetic and epigenetic mechanisms related to endometriosis.  However, there are two major sections discussing about steroidogenic pathways and SF-1.  I hope the authors may think it over either to modify the title or link these sections more closely to the genetics and epigenetics. 
  2. The second sentence of Introduction (lines 33-35) is controversial. Endometriosis is characterized by presence of ectopic endometrial tissue outside the uterus.  Presence of active uterine mucosa (glands and stroma) defines the utopic endometrium.  Authors need to modify. 
  3. I like Figure 1, which summarized the majority pathogenesis of endometriosis. However, as I am aware of that fallopian tube also contributes to the formation of endometriosis, particularly endometriosis in the ovary (Yuan et al., Modern Pathology 2014, volume 27, pages1154–1162).  This represents an important contribution to the endometriosis field, which may be promising in thinking of future direction of prevention and therapeutics.  I would like to suggest the authors give brief coverage about this and include into the Figure 1.  Within Figure 1, you may combine “fetal origin and Embryonic” in two separate cells into one “Fetal/embryonic origin”. 
  4. Good review articles typically contain commentary summary in each section. The authors have give such statements in many sections except section 2.1, the Family studies in Endometriosis. 
  5. In the sectin of steroidogenic pathway, lines 281 to 292, authors repeated twice with Sapkota et al (reference #17) twice in two consecutive paragraphs. This does not read well and it is not necessary to cite twice with the same language.  Modification is required. 
  6. The sentence (lines 353 to 355) has a grammar error and does not sound complete. It should also include cellular differentiation after mitosis and/or meiosis process.  Authors should consider to revise. 
  7. There are several typos including sentences in lines 410, 509, 567 etc. These should be an easy fix.

Reviewer 2 Report

The aim, the methodology and the discussion of the study are well structured and written; the conclusions are supported by the analysis of the data presented. The paper is high quality and suitable for the field of the journal.

Minor points which I think the authors may wish to address are:

Inclusion and exclusion criteria set the boundaries for a review: Did they exclude case reports? Moreover, did they exclude Whole exome sequencing studies or fine mapping studies?

Line 24-24: this biomarker can further inform us about the effectiveness of treatment and the stage of the disease.

Line 48: On MRI superficial lesions are not visible because they are tiny and flat. I would rather mention deep lesions.

Figure 1: Which is the difference between fetal / embryonic origin?

line 163-165: comment: Non-coded areas, which cover about 95% of the human genome, are important sequences for the transcription and translational function of genes. For example, if a polymorphism is in an intron or a promoter, it could affect the effective binding of a transcription factor, resulting in the expression of the corresponding protein at reduced levels.

Authors can further mention studies performed with classical PCR, such as IL-16, also known as the chemotactic factor of lymphocytes that plays a key role in various immune and inflammatory reactions. It activates T-lymphocytes, leading to the secretion of several pro-inflammatory cytokines, resulting in the survival of the ectopic endometrial tissue in the peritoneal cavity.

All figures effectively and efficiently convey this complex detailed information.
